# Catheter ablation of atrial arrhythmias in cardiac amyloidosis: Impact on heart failure and mortality

**Philippe Maury**[1,2]*, **Kevin Sanchis**[1], **Kamila Djouadi**[3], **Eve Cariou**[1], **Hubert Delasnerie**[1], **Serge Boveda**[4], **Pauline Fournier**[1], **Romain Itier**[1], **Pierre Mondoly**[1], **Quentin Voglimacci-Stephanopoli**[1], **Maxime Beneyto**[1], **Tarvinder S. Dhanjal**[5], **Anne Rollin**[1], **Thibaud Damy**[3], **Olivier Lairez**[1], **Nicolas Lellouche**[3]

**1** Department of Cardiology, University Hospital Toulouse, Toulouse, France, **2** I2MC, INSERM UMR 1297, Toulouse, France, **3** Department of Cardiology, University Hospital Henri Mondor, Creteil, France, **4** Clinique Pasteur, Toulouse, France, **5** Department of Cardiology, University of Warwick, Coventry, United Kingdom

* mauryjphil@hotmail.com

**Data Availability Statement:** All relevant data are within the paper and its Supporting Information files.

**Funding:** The authors received no specific funding for this work.

## Abstract

### Background

Atrial arrhythmias (AA) commonly affect patients with cardiac amyloidosis (CA) and are a contributing risk factor for the development of heart failure (HF). This study sought to investigate the long-term efficacy and impact of catheter ablation on HF progression in patients with CA and AA.

### Methods

Thirty-one patients with CA and AA undergoing catheter ablation were retrospectively included (transthyretin—ATTR CA 61% and light chain—AL CA 39%). AA subtypes included atrial fibrillation (AFib) in 22 (paroxysmal in 10 and persistent in 12), atrial flutter (AFl) in 17 and atrial tachycardia (AT) in 11 patients. Long-term AA recurrence rates were evaluated along with the impact of sinus rhythm (SR) maintenance on HF and mortality.

### Results

AA recurrence was observed in 14 patients (45%) at a median of 3.5 months (AFib n = 8, AT n = 6, AFl = 0). Post-cardioversion, medical therapy or catheter ablation, 10 patients (32%) remained in permanent AA. Over a median follow-up of 19 months, all-cause mortality was 39% (n = 12): 3 with end-stage HF, 5 due to late complications of CA, 1 sudden cardiac death, 1 stroke, 1 COVID 19 (and one unknown). With maintenance of SR following catheter ablation, significant reductions in serum creatinine and natriuretic peptide levels were observed with improvements in NYHA class. Two patients required hospitalization for HF in the SR maintenance cohort compared to 5 patients in the AA recurrence cohort (p = 0.1). All 3 patients with deaths secondary to HF had AA recurrence compared to 11 out of the 28 patients whom were long-term survivors or deaths not related to HF (p = 0.04). All-cause mortality was not associated with AA recurrence.

**Competing interests:** The authors have declared that no competing interests exist.

## Conclusion

This study demonstrates moderate long-term efficacy of SR maintenance with catheter ablation for AA in patients with CA. Improvements in clinical and biological status with positive trends in HF mortality are observed if SR can be maintained.

## Introduction

Cardiac amyloidosis (CA) results from the deposition of insoluble misfolded fibrous proteins within the myocardium leading to progressive heart failure (HF), stereotypically with preserved ejection fraction and a restrictive pattern [1]. In addition to increasing age, elevated left ventricular (LV) filling pressures and dilatation of the left atrium, amyloid deposition within the atrial myocardium (amyloid atriopathy)has been proposed to be a contributing factor to the high prevalence of atrial arrhythmias (AA) observed in CA [2,3].

AA are associated with rapid ventricular rates and loss of sinus rhythm (SR), with subsequent atrioventricular desynchronization, which can potentially result in hemodynamic destabilization and heart failure in patients with CA [4]. In addition, diastolic dysfunction as seen in CA can by exacerbated by loss of atrial systole and reduced early diastolic filling in AA [5]. However, the benefits of SR maintenance may be limited in patients with CA, due to the restrictive diastolic dysfunction in this population [6]. Previous reports suggest that the presence of atrial fibrillation (AFib) does not impact all-cause mortality in CA patients [7, 8]. Our group has previously shown that AFib was not shown to have any significant impact on cardiovascular mortality, whatever the subtypes of CA or AFib [9].

Since cardiac output in CA is highly dependent on heart rate because of altered diastolic filling and since CA is associated with frequent conduction disturbances, rate control is challenging because of the vasodilator, potentially toxic, negative inotropic, dromotropic and especially chronotropic effects of drugs [2, 6]. Due to before mentioned reasons, amiodarone is the only anti-arrhythmic drug used in this population, but it has been shown to be ineffective in maintaining SR in majority of the cases [10]. However, catheter ablation for AFib has been demonstrated to significantly improve HF symptoms and progression [11, 12] with a mortality benefit [13] compared to drug therapy in patients with HF and impaired LV ejection fraction (LVEF). However, in the limited case series reported to-date, there is considerable variation in the long-term efficacy of AFib ablation in patients with CA [14–17]. Catheter ablation is however a first-line and curative treatment for typical atrial flutter (AFl).

The aim of this study was to explore the efficacy and long-term impact on HF status and mortality of catheter ablation in CA patients complicated with AA.

## Methods

### Study population

All patients with CA undergoing catheter ablation for AA at Toulouse and Mondor Paris University Hospitals from 2014 to 2021 were retrospectively included.

Diagnosis of CA was defined by echocardiogram parameters (interventricular septum diameter > 12 mm in the absence of any other cause of ventricular hypertrophy [18] and apex-to-base gradient in regional longitudinal strain (LS) > 1.0 [19]) in presence of systemic amyloidosis (for light chain amyloidosis–AL) or grade 2–3 cardiac uptake on a bone $^{99m}$Tc-hydroxymethylene-diphosphonate scintigraphy in patients without monoclonal protein (for

transthyretin amyloidosis–ATTR) [20]. Systemic amyloidosis was defined by histological documentation of Congo Red staining and apple-green birefringence under cross-polarized light in at least one organ [21]. For patients with ATTR, familial ATTR was diagnosed after identification of a pathogenic variant on transthyretin gene by DNA analysis.

### Electrophysiology and ablation

AA consisted of paroxysmal or persistent AFib, AFl or atrial tachycardia (AT). History of AA, AADs, electrical cardioversions, pacemaker or implantable cardioverter-defibrillator (ICD) implantation and sick sinus syndrome was assessed.

Catheter ablation was performed using standard techniques and endpoints [22] using radio-frequency energy. Diagnosis of cavotricuspid isthmus (CTI) dependent AFl was made by 12-lead ECG pattern and entrainment maneuvers. Procedural endpoint was complete bidirectional CTI block. AT was defined by regular AA not related to CTI-dependent flutter and was ablated using 3D electro-anatomic mapping (EAM). Scar surfaces (low voltage atrial myocardium) were exported from EAM voltage maps into Matlab ™ with analysis performed comparing to total atrial surfaces. Endpoint was termination of the AT with either reentry isthmus ablation or delivering RF to the focal AT site, resulting in non-inducibility. For AFib ablation, complete pulmonary vein isolation (PVI) was performed in all cases, with additional linear lines (roof and mitral isthmus) and/or defragmentation of fractionated areas, resulting in SR or conversion to AT which was subsequently targeted. Otherwise, external DC shock was performed with confirmation of PVI and linear lesions were assessed and completed for block if needed.

### Clinical endpoint and follow-up

Follow-up was assessed by the consulting physician/cardiologist, patient or family for the clinical endpoints of all-cause mortality, HF, relapse of AA and hospitalization. Weight, NYHA class, serum creatinine and NT-pro BNP levels as well as echocardiographic parameters (left atrial volume, systolic pulmonary arterial pressure) were assessed at baseline prior to ablation and post-ablation in SR.

Signed informed consent was obtained from all patients. In accordance with French ethics and regulatory law, this retrospective study was approved by reference methodology of the French National Commission for Informatics and Liberties (CNIL). The study was registered at Toulouse University Hospital and covered by the MR-004 (CNIL number: 2206723 v 0).

### Statistical analysis

Continuous variables were expressed as median and inter-quartile ranges (IQR) and compared with Mann Whitney test (unpaired comparisons) or Wilcoxon test (paired comparisons). Nominal variables were expressed as numbers and percentages and compared with Fisher exact test. Differences were considered statistically significant for p values $< 0.05$.

## Results

### Patient population

Thirty-one patients with CA and AA undergoing catheter ablation were retrospectively included (26 males, 74 yr, IQR 15). Baseline characteristics are shown in Table 1. Nineteen presented with ATTR CA (61%, familial in 4, all 19 treated with tafamidis), while the remaining

**Table 1. Baseline patient characteristics.**

| | |
|---|---|
| Gender | 26 males (84%) |
| Age (yr) | 74 (IQR 15) |
| Type of amyloidosis | 19 ATTR (61%) <br> 12 AL (39%) |
| Hypertension | 22 (71%) |
| Diabetes mellitus | 7 (23%) |
| Reduced LVEF < 50% | 11 (35%) (median 41 IQR 12) |
| Previous congestive heart failure | 19 (61%) |
| Associated ischemic cardiomyopathy | 9 (29%) |
| LVEF (whole population) (%) | 53 (IQR 14) |
| NYHA score | 2 (IQR 1) |
| CHA2DS2VaSC score | 3 (IQR 2) |
| European score (AL-CA) <br> NAC score (ATTR-CA) | II 2/9 IIIa 6/9 III b 1/9 <br> I 5/18 II 5/18 III 8/18 |
| Previous amiodarone therapy | 26 (84%) |
| Previous other anti-arrhythmic drugs | 12 (39%) |
| Current amiodarone therapy | 12 (39%) |
| Beta-blockers | 14 (45%) |
| Calcium channel blockers | 2 (6%) |
| Digoxin | 2 (6%) |
| Oral anticoagulation | 30 (97%) |
| Anticoagulant drugs | Vitamin K Antagonists 15 (50%) <br> Apixaban 9 (30%) <br> Rivaroxaban 4 (13%) <br> Dabigatran 2 (7%) |
| Pacemaker/ICD | 14 (45%) / 12 (39%) |

ICD: Implantable cardioverter-defibrillator; LVEF: left ventricular ejection fraction; NYHA: New York Heart Association

12 patients (39%) suffered with AL CA (myeloma, 1; lymphoma, 1; prostate neoplasm, 1; lambda light chains,9; kappa chains,2; unknown, 1). Nine patients (29%) had associated ischemic heart disease, 19 (61%) presented with previous HF and reduced LVEF was noted in 9 (29%). Four patients (13%) had experienced an embolic event, while 3 (10%) had presented with hemorrhagic complications during oral anticoagulation.

CA had been diagnosed at a mean age of 71 yr (IQR 11) and AA were first detected at a mean age of 71 yr (IQR 14): pre-CA diagnosis in 11 patients (30 months before, IQR 90), concomitantly in 8 and post-CA diagnosis in 12 (10 months after, IQR 25).

All 31 patients underwent catheter ablation for AA: 22 (71%) for Afib (paroxysmal in 10 and persistent in 12), 17 (55%) for CTI-dependent AF and 11 (35%) for AT. Previous electrical cardioversion had been performed in 14 patients (45%). None had experienced sick sinus syndrome. Two patients had a history of prior catheter ablation (for AFl) and 3 patients had previously undergone atrio-ventricular node ablation. Fourteen and 12 patients had previously undergone pacemaker or ICD implantation, respectively. None had undergone closure of the left atrial appendage.

Mean heart rate for all patients was 78 bpm (IQR 21), mean PR interval was 215 msec (IQR 60) for patients presenting in SR, and QRS duration was 140 msec (IQR 70).

Patients with AL CA presented at a younger age with lower CHA$_2$DS$_2$VaSC scores and more paroxysmal AF and HF compared with ATTR patients.

## Ablation

Ablation was performed at a mean of 18 months (IQR 41) following the first documentation of AA (excluding the 2 cases with previous AFl ablation). The median scar surface area was 85 cm$^2$ (IQR 70) representing 75% of the total atrial surface mapped (IQR 36). Acute procedural success was achieved in all cases with all patients being discharged in SR. Seven patients (23%) were prescribed AADs at discharge (amiodarone, 5; flecainide and sotalol,1).

## Follow-up

Eleven patients (35%) were receiving long-term class 1 or class 3 AADs. Recurrences of any AA following ablation were observed in 14 patients (45%) at a median of 3.5 months (IQR 18).

- AFib recurred in eight patients and four underwent electrical cardioversion (AFl ablated in one patient two months later). Three were prescribed long-term AADs.

- AT recurred in 6 patients: four underwent electrical cardioversion and one a further ablation procedure 4 months after the initial procedure. Four were prescribed long-term AADs.

- AFl did not recur in any patient.

Of the 17 patients with SR maintenance following ablation (55%), eight had undergone ablation for AFl and four were prescribed long-term AADs.

AA recurrences were not associated with age, gender, type of CA, hypertension, diabetes, HF, LVEF, NHYA class, ischemic cardiomyopathy, presence of pacemaker/ICD or long-term AADs, but were less frequent in patients treated with amiodarone (25% vs 58%, p = 0.07) and less frequent in patients without beta-blockers (29 vs 64%, p = 0.05) at the time of the ablation procedure. There were no significant differences in the age at CA diagnosis or in the duration of CA diagnosis between patients with or without AA recurrences.

At final follow-up, 10 patients (32%) remained in AA and 21 (68%) maintained SR, as shown in Fig 1.

Over a median follow-up of 19 months (IQR 26), all-cause mortality was 39% (12 patients): three patients died from end-stage HF, one from sudden cardiac death, one from stroke, while late noncardiac complications of amyloidosis were the cause of death in five patients (pneumopathy in two, degradation of general condition in two and sepsis in one). One patient died from Covid19 infection and cause of death was unknown for one patient. There were no differences in AA recurrence, mortality or HF hospitalizations between ATTR CA and AL CA. There was no significant difference in mortality between ICD-implanted and non-implanted patients (25% vs 47%, p = ns)

## Impact of ablation on outcomes

Over a median follow-up of 9 months (IQR 17), serum creatinine and natriuretic peptides levels significantly decreased with improvements in NYHA class in patients remaining in sinus rhythm, without a significant change in weight. Left atrial dimension and pulmonary arterial pressure decreased after ablation in patients in sinus rhythm although not significantly (see Table 2).

There was a trend toward less hospitalizations for HF in patients remaining in SR over the follow-up period (2/21) compared to patients with AA recurrence (5/10) (p = 0.1).

All 3 patients with deaths caused by intractable HF had AA recurrence compared to 11 out of the remaining 28 long-term survivors (or dead but not related to HF) (p = 0.04). All-cause mortality was not associated with AA recurrence (5/17 wo AA recurrence vs 7/14 with AA recurrence, p = 0.2), and mortality in patients with permanent AA (5/10) was not significantly higher compared to patients remaining in SR (7/21) (p = 0.3).

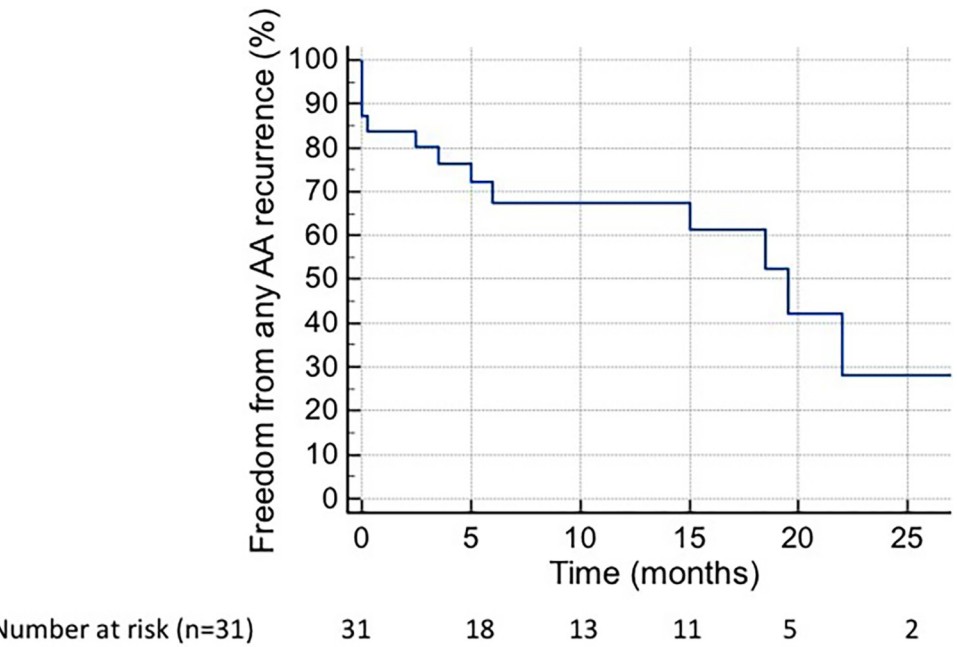

**Fig 1. Kaplan-Meier curve of freedom from atrial AF/atrial flutter recurrence following ablation.** There was 30% recurrences at one year and 70% at 24 months (not including redo procedures). AFib: Atrial Fibrillation.

## Discussion

This study demonstrates that catheter ablation of AA in CA, results in modest long-term success rates, with improvements in clinical and biological status and positive trends in cardiovascular morbidity if SR can be maintained.

Prevalence of AFib in patients with CA is widely variable amongst studies [7, 23], with reported rates ranging from 9% to 50% in AL amyloidosis and up to 70% in wt-ATTR amyloidosis [9, 10]. The prevalence of AFib increased along with the NYHA stage in the study by Longhi et al. [7]. A 9% prevalence of AFl in CA has been reported [9] whilst data on the prevalence of AT is lacking. Surprisingly, AFib in patients with CA was not shown to impact all-cause mortality. Indeed the present study and other previous reports did not find that AFib significantly altered survival in patients with CA, whatever the type of amyloidosis [9, 10]. The high prevalence and recurrence rates of AFib and the poor prognosis of CA alone likely explain the lack of additional impact of AFl on prognosis. These results question the role of rhythm control for the management of AFib in CA.

Most of the studies investigating CA and AFib have only reported outcomes of medically treated patients with or without electrical cardioversion, without including patients

**Table 2. NYHA class, biological and echocardiogram changes pre- and post-ablation (while in SR).**

|  | Before ablation | After ablation | p value |
| --- | --- | --- | --- |
| Creatinine (μmol/l) | 127 (IQR 56) | 108 (IQR 59) | 0.003 |
| NT-proBNP (pg/mL) | 3212 (IQR 4041) | 1915 (IQR 2379) | 0.01 |
| Weight (kg) | 81 (IQR 19) | 77 (IQR 16) | 0.16 |
| NYHA class | 3 (IQR 1) | 2 (IQR 0) | 0.0009 |
| Left atrial volume (ml/m$^2$) | 55 (IQR 22) | 46 (IQR 24) | 0.6 |
| Systolic pulmonary arterial pressure (mmHg) | 42 (IQR 14) | 37 (IQR 11) | 0.26 |

undergoing catheter ablation, despite ablation therapy being shown to be superior to AAD therapy [24–26] or a pace and atrioventricular node ablation strategy [27].

There are only a few studies including limited numbers of patients with catheter ablation of AA in CA [10, 14–16]. In the first study including 13 patients and various arrhythmias, one-year and 3-year recurrence-free survival rates were 75% and 60%, respectively [15]. However, in another short series of 7 CA patients undergoing left atrial mapping and ablation for persistent AT/AFib, the recurrence rate at one year was high and significantly greater compared to controls (83% vs 25%) [14]. More recently, in 24 patients with ATTR-CA and AFib undergoing ablation, recurrences occurred in 58% at 40 months follow-up, significantly more in end-stage CA (90%) and less in patients receiving tafamidis [10]. Arrhythmia-free survival after catheter ablation of various AA was 40% at 1 year and 20% at 2 years in another recent short series of 10 CA patients [16]. Thus, rather poor long-term success rates have been reported, significantly lower compared to standard AFib populations, likely reflecting both the diversity of the presenting AA and the challenging adverse atrial remodelling that is associated with CA. This is in comparison to other clinical situations, where long-term success rates of Afib ablation range between 50% and 80% [28]. Moreover, the positive impact of sinus rhythm maintenance with Afib ablation in heart failure is well established [13].

Both randomized and non-randomized clinical trials comparing rhythm control and rate control strategies in patients with CA are lacking. In this population where cardiac output is low and dependent on elevated heart rates [29], the benefit of a rhythm control strategy over a rate control strategy has still to be investigated. Although the strategy of rhythm control does not seem to impact survival in retrospective studies, it is likely however, that maintenance of SR could improve symptoms and limit the progression of HF. It is plausible that restoring atrial systole may improve LV filling and cardiac output hence reduce symptoms in this population of patients with impaired diastolic ventricular function. A previous study has shown that AFib in CA was strongly associated with HF [7]. Our group has previously reported that patients without a previous history of AFib and including those who did not experience AFib during follow-up were less symptomatic [9]. However this may simply suggest that the more severe the CA the higher the risk of AFib occurrence.

A previous study investigating AA ablation in CA reported some improvement in NYHA class after ablation in 2/3 of patients but with 23% mortality [15]. Mortality was significantly lower after ablation compared to matched non-ablated patients in a further study (29 vs 70%) and ablation was associated with a significant reduction in the frequency of hospitalization for HF or arrhythmias [10]. Ablation seems to be more effective when performed in the earlier stages of the disease [10, 16].

However, there have been no prior reports on changes in hemodynamic and clinical status following AA ablation in CA. In this study, we demonstrated that ablation of AA in CA, had modest long-term success rates but may translate into some improvements in clinical and biological status, with trends towards a reduction in cardiovascular mortality/morbidity if SR can be maintained. The impact on mortality may have been significant with a larger population. Thus, performing timely catheter ablation for any atrial arrhythmia in patients with CA should be considered.

## Limitations

This is a retrospective study sharing all the associated limitations and bias. The cohort comprised patients referred to our tertiary center for initial management and evaluation, thus this cohort may not match the standard population of patients with CA and AA.

Presence of ICD or previous atrio-ventricular node ablation may have contributed to morbidity or mortality outcomes, however no differences in mortality between implanted and

non-implanted patients were observed, and the few patients with prior atrio-ventricular node ablation manifested congestive heart failure despite ventricular rate control prior to AA ablation.

Left atrial strain has been reported to correlate with the risk of AFib, independent of the left atrial size [3], and could be helpful to select responders to cardioversion or ablation. Furthermore, several studies have shown improved SR maintenance rates when the left atrial strain is preserved [27, 30]. Left atrial strain was not systematically measured in this study.

## Conclusion

Ablation of AA in CA, although a challenging task with modest long-term success rates, may translate into some improvements in clinical and biological status and positive trends in cardiovascular mortality if SR can be maintained.

## Supporting information

**S1 Data.**
(XLSM)

## Author Contributions

**Conceptualization:** Philippe Maury, Nicolas Lellouche.

**Data curation:** Philippe Maury, Kevin Sanchis, Kamila Djouadi, Eve Cariou, Hubert Delasnerie, Serge Boveda, Pauline Fournier, Romain Itier, Pierre Mondoly, Quentin Voglimacci-Stephanopoli, Maxime Beneyto, Anne Rollin, Olivier Lairez, Nicolas Lellouche.

**Formal analysis:** Philippe Maury.

**Investigation:** Philippe Maury, Kamila Djouadi.

**Methodology:** Philippe Maury.

**Supervision:** Philippe Maury, Kamila Djouadi, Eve Cariou, Hubert Delasnerie, Serge Boveda, Pauline Fournier, Romain Itier, Pierre Mondoly, Quentin Voglimacci-Stephanopoli, Maxime Beneyto, Anne Rollin, Thibaud Damy, Olivier Lairez, Nicolas Lellouche.

**Validation:** Philippe Maury, Kevin Sanchis, Eve Cariou, Hubert Delasnerie, Pauline Fournier, Romain Itier, Quentin Voglimacci-Stephanopoli, Maxime Beneyto, Anne Rollin, Thibaud Damy, Nicolas Lellouche.

**Visualization:** Philippe Maury.

**Writing – original draft:** Philippe Maury.

**Writing – review & editing:** Philippe Maury, Tarvinder S. Dhanjal.

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
