## [Decision Letter · Decision Letter 0]

28 Nov 2022

PONE-D-22-29848Catheter Ablation of Atrial Arrhythmias in Cardiac Amyloidosis

Impact on Heart Failure and MortalityPLOS ONE

Dear Dr. Maury,

Thank you for submitting your manuscript to PLOS ONE. After careful consideration, we feel that it has merit but does not fully meet PLOS ONE’s publication criteria as it currently stands. Therefore, we invite you to submit a revised version of the manuscript that addresses the points raised during the review process.

Please submit your revised manuscript by Jan 12, 2023, 11:59PM. If you will need more time than this to complete your revisions, please reply to this message or contact the journal office at plosone@plos.org. Please include the following items when submitting your revised manuscript:A rebuttal letter that responds to each point raised by the academic editor and reviewer(s). You should upload this letter as a separate file labeled 'Response to Reviewers'.A marked-up copy of your manuscript that highlights changes made to the original version. You should upload this as a separate file labeled 'Revised Manuscript with Track Changes'.An unmarked version of your revised paper without tracked changes. You should upload this as a separate file labeled 'Manuscript'.

We look forward to receiving your revised manuscript.

Kind regards,

Daniel A. Morris, M.D

Academic Editor

PLOS ONE

“No funding”

“No competing interest”

6. Please include your tables as part of your main manuscript and remove the individual files. Please note that supplementary tables (should remain/ be uploaded) as separate "supporting information" files

Additional Editor Comments:

The aim of this study is clinically relevant. However, there are some major limitations in this study that should be addressed.

1) Small sample size: just 22 patients had AF, which is the more important event/variable to analyze. Hence, the authors should increment significant the sample size of the study, including at least 200 patients in order to avoid bias and to increment the clinically relevance of this study.

2) Subgroup analysis: Patients with Al-Amyloidosis and ATTR-Amyloidosis should be analyzed separately, since these subtypes of amyloidosis have significantly different outcomes and treatment.

3) The primary endpoint of this study should be only recurrence of AF at 1 and 2 years of follow-up.

Reviewers' comments:

Reviewer's Responses to Questions

**Comments to the Author**

1. Is the manuscript technically sound, and do the data support the conclusions?

Reviewer #1: Yes

Reviewer #2: Yes

2. Has the statistical analysis been performed appropriately and rigorously? 

Reviewer #1: N/A

Reviewer #2: Yes

3. Have the authors made all data underlying the findings in their manuscript fully available?

Reviewer #1: No

Reviewer #2: Yes

4. Is the manuscript presented in an intelligible fashion and written in standard English?

Reviewer #1: Yes

Reviewer #2: Yes

5. Review Comments to the Author

Reviewer #1: In the current paper, the authors reported their experience about the impact of catheter ablation on outcomes in patients with amyloidosis. The paper was well written in general. Please include more data about the catheter ablation procedure regarding the left atrial scarring in the study group.

Reviewer #2: The authors report on outcome post catheter ablation for atrial arrhytmias (AT, Aflutter or AF) in a patient series comprising 31 patients with cardiac amyloidosi.

Cardiac amyloidosis is an infiltrative disease with poor outcome, bacause of development of advanced heart failure leadingt o death.

Although the presented case series is limited with regard to the number of patients, the study is interesting and one of the rare reports on this topic. The study demonstartes the potential for improved hemodynamics, NAHY class and BNP levels, in those CA-patients in whom sinus rhythm can be maintained by combination of cathetr ablation and antiarrhythmic drug therapy. Notably, betablocker treatment was not beneficial in this patient cohort with restrictive CMP.

---

## [Author Response · Author response to Decision Letter 0]

7 Jan 2023

PONE-D-22-29848

Catheter Ablation of Atrial Arrhythmias in Cardiac Amyloidosis

Impact on Heart Failure and Mortality

PLOS ONE

Additional Editor Comments:

The aim of this study is clinically relevant. However, there are some major limitations in this study that should be addressed.

1) Small sample size: just 22 patients had AF, which is the more important event/variable to analyze. Hence, the authors should increment significant the sample size of the study, including at least 200 patients in order to avoid bias and to increment the clinically relevance of this study.

Answer : this is the biggest series of AF ablation in cardiac amyloidosis to date. Probably far less than 50 patients have been ablated worldwide, because it is absolutely not the experience of most centers. Thus collecting 200 patients is simply impossible at yet. We feel that this series is large enough to draw some conclusions and anyway will be impossible to increment.

2) Subgroup analysis: Patients with Al-Amyloidosis and ATTR-Amyloidosis should be analyzed separately, since these subtypes of amyloidosis have significantly different outcomes and treatment.

Answer : this has been done :

- « There was no significant difference between patients with ATTR and AL CA, except more paroxysmal AF and congestive heart failure in AL CA, with a younger age and lower CHA2DS2VaSC score. » page 7 line 13

- « Recurrences were not associated with age, gender, type of CA … » page 8 line 10

- « There was no difference in recurrence, mortality or hospitalizations for heart failure between ATTR CA and AL CA. » page 8 line 22

- « All the three patients with end-stage HF leading to death (all with ATTR AF) had recurrent arrhythmias compared to .. » page 9 line 8

3) The primary endpoint of this study should be only recurrence of AF at 1 and 2 years of follow-up.

Answer : the primary endpoint was to assess the long-term impact of catheter ablation for atrial arrhythmias in cardiac amyloidosis, and especially if this will translate in better hemodynamics and less congestive heatt failure and eventually in prognosis. This was not a study devoted to crude results of ablation and about technique and so on.

Moreover, in the results, we mention the long terme results of ablation : « At final follow-up, 10 patients (32%) were in AA and 21 (68%) remained in SR over a median follow-up of 19 months » and the survival curve (free from recurreces) also is shown. So we feel that these informations are already present.

Reviewers' comments:

Reviewer #1: In the current paper, the authors reported their experience about the impact of catheter ablation on outcomes in patients with amyloidosis. The paper was well written in general. Please include more data about the catheter ablation procedure regarding the left atrial scarring in the study group

Answer : the paper was not dedicated to precise description of ablation procedures in CA patients, but datas on left atrial scarring in patients with atrial fibrillation has been added : 

« Scar surfaces (atrial tissues with less than normal voltage) were retrieved from 3D voltage maps using Mathlab ™ program and correlated to total atrial surfaces. » page 5 line 2

« Median scar surface was 85 cm2 (IQR 70) representing 75 % of total atrial surface (IQR 36). » page 7 line 18

Reviewer #2: The authors report on outcome post catheter ablation for atrial arrhytmias (AT, Aflutter or AF) in a patient series comprising 31 patients with cardiac amyloidosi.

Cardiac amyloidosis is an infiltrative disease with poor outcome, bacause of development of advanced heart failure leadingt o death.

Although the presented case series is limited with regard to the number of patients, the study is interesting and one of the rare reports on this topic. The study demonstartes the potential for improved hemodynamics, NAHY class and BNP levels, in those CA-patients in whom sinus rhythm can be maintained by combination of cathetr ablation and antiarrhythmic drug therapy. Notably, betablocker treatment was not beneficial in this patient cohort with restrictive CMP.

Answer : the thank the reviewers for these nice comments

---

## [Editor Report · Decision Letter 1]

13 Jan 2023

PONE-D-22-29848R1

Catheter Ablation of Atrial Arrhythmias in Cardiac Amyloidosis. Impact on Heart Failure and Mortality

PLOS ONE

Dear Dr. Maury,

Thank you for submitting your manuscript to PLOS ONE. After careful consideration, we have decided that your manuscript does not meet our criteria for publication and must therefore be rejected.

Specifically, as it has been stated in the first revision, this small, descriptive, and retrospective report has serious limitations and major biases such as very small sample size (only 22 patients had AF), lack of subgroup analysis (Al- and ATTR-Amyloidosis), and lack of primary endpoint analysis at 1 and 2 years (i.e., recurrence of AF at 1 and 2 years of follow-up). 

I am sorry that we cannot be more positive on this occasion, but hope that you appreciate the reasons for this decision.

Kind regards,

Daniel A. Morris, M.D

Academic Editor

PLOS ONE

---

## [Author Response · Author response to Decision Letter 1]

12 Feb 2023

Catheter Ablation of Atrial Arrhythmias in Cardiac Amyloidosis

Impact on Heart Failure and Mortality

PLOS ONE

PONE-D-22-29848R1

Details on the revisions carried out on the manuscript since its original submission

We already resubmitted a revised version end 2022 according to the editor’s remarks and concerns and reviewer comments, including the following changes. Additional editor’s comments were answered onset 2023, which were in fact fundamentally the same ones, thus our answers were also similar. This point by point response gathers all the previously answers and comments to the editor’s / reviewers concerns.

Answer to the Editor’s Comments:

The aim of this study is clinically relevant. However, there are some major limitations in this study that should be addressed.

1) Small sample size: just 22 patients had AF, which is the more important event/variable to analyze. Hence, the authors should increment significant the sample size of the study, including at least 200 patients in order to avoid bias and to increment the clinically relevance of this study.

Answer : this is the biggest series of AF ablation in cardiac amyloidosis to date. Probably far less than 50 patients have been ablated worldwide, because it is absolutely not the experience of most centers. Thus collecting 200 patients is simply impossible at yet. We feel that this series is large enough to draw some conclusions and anyway will be impossible to significantly increment currently at the point the editors wished.

2) Subgroup analysis: Patients with Al-Amyloidosis and ATTR-Amyloidosis should be analyzed separately, since these subtypes of amyloidosis have significantly different outcomes and treatment.

Answer : this has been done :

- « There was no significant difference between patients with ATTR and AL CA, except more paroxysmal AF and congestive heart failure in AL CA, with a younger age and lower CHA2DS2VaSC score. » page 8 line 10

- « Recurrences were not associated with age, gender, type of CA … » page 9 line 7

- « There was no difference in recurrence, mortality or hospitalizations for heart failure between ATTR CA and AL CA. » page 9 line 18

- « All the three patients with end-stage HF leading to death (all with ATTR AF) had recurrent arrhythmias compared to .. » page 10 line 8

3) The primary endpoint of this study should be only recurrence of AF at 1 and 2 years of follow-up.

Answer : the primary endpoint was not to assess the long term success rate of ablation, but rather if this will translate in better hemodynamics and less congestive heart failure and eventually in prognosis. 

This was not a study devoted to crude results of ablation and about technique and so on. Thus, the primary endpoint was to assess the long-term impact of catheter ablation for atrial arrhythmias in cardiac amyloidosis regarding hearte failure and prognosis. 

Anyway, in the results, long term results of ablation are mentioned: 

« At final follow-up, 10 patients (32%) were in AA and 21 (68%) remained in SR over a median follow-up of 19 months » and the survival curve (free from recurrences) also is shown. On the survival curve one can see that there was 30% recurrences at one year and 70 % at 24 months (not including redo procedures). This has been added in the figure legend. So we feel that these informations are in fact more clearly present.

Answers to the Reviewers' comments:

Reviewer #1: In the current paper, the authors reported their experience about the impact of catheter ablation on outcomes in patients with amyloidosis. The paper was well written in general. Please include more data about the catheter ablation procedure regarding the left atrial scarring in the study group

Answer : the paper was not dedicated to precise description of ablation procedures in CA patients, but datas on left atrial scarring in patients with atrial fibrillation has been added : 

« Scar surfaces (atrial tissues with less than normal voltage) were retrieved from 3D voltage maps using Mathlab ™ program and correlated to total atrial surfaces. » page 5 line 2

« Median scar surface was 85 cm2 (IQR 70) representing 75 % of total atrial surface (IQR 36). » page 8 line 15

Reviewer #2: The authors report on outcome post catheter ablation for atrial arrhytmias (AT, Aflutter or AF) in a patient series comprising 31 patients with cardiac amyloidosi.

Cardiac amyloidosis is an infiltrative disease with poor outcome, bacause of development of advanced heart failure leadingt o death.

Although the presented case series is limited with regard to the number of patients, the study is interesting and one of the rare reports on this topic. The study demonstartes the potential for improved hemodynamics, NAHY class and BNP levels, in those CA-patients in whom sinus rhythm can be maintained by combination of cathetr ablation and antiarrhythmic drug therapy. Notably, betablocker treatment was not beneficial in this patient cohort with restrictive CMP.

Answer : the thank the reviewers for these nice comments

---

## [Decision Letter · Decision Letter 2]

18 Sep 2023

PONE-D-22-29848R2Catheter Ablation of Atrial Arrhythmias in Cardiac Amyloidosis. Impact on Heart Failure and MortalityPLOS ONE

Dear Dr. Maury,

Thank you for submitting your manuscript to PLOS ONE. After careful consideration, we feel that it has merit but does not fully meet PLOS ONE’s publication criteria as it currently stands. Therefore, we invite you to submit a revised version of the manuscript that addresses the points raised during the review process.

**1. **There are minor grammatical errors in the abstract section. In methods section: “ AA were atrial fibrillation..” this statement needs better grammatical construction. Also in this section authors mention “ Atrial common flutter” , do you mean “ Atrial flutter” 

**2.** Its not clear looking at the abstract about what the main aims of the study are. In the methods section—may be we should explain that we compared outcomes before and after ablation etc? to assess long term efficacy of ablation etc

       Throughout rest of the article, there are statements which need grammatical correction such as

  “AA undergoing ablation were AFib in 22 (71%) (paroxysmal in 10 and persistent in 12), CTI-dependent AF in 17 (55%) and AT in 11 (35%)”---- better statement----   ‘ *Among patients undergoing ablation for arrythmia, 71%  had afib, 55% had CTI-dependent afib and 11% had AT” etc*

*“*A previous history of catheter ablation was present in only two cases (for AF)” –  correct grammar “ only 2 cases had a hx of catheter ablation prior to our study.

“When comparing before and after ablation while in SR (median 9 months later, IQR 17), creatinine and natriuretic peptides levels significantly decreased as well as NYHA class, without relevant change in weight, while left atrial dimensions and pulmonary arterial pressures, although decreased, did not significantly change (see table 2)”--   Full stop after “change in weight”. Then last statement should start ( my suggestion)

may be get assistance of professional grammatical correction services

**3.** How do we know that this study shows long term efficacy of ablation in CA patients?

Is ablation in CA patients better or worse than ablation in patients without CA? Whats the control population to compare with?. How do we know that history of amyloidosis interacting/or not interacting with ablation outcomes

If we compared effects of ablation between patients who had cardiac amyloidosis and patients who did not have cardiac amyloidosis: then significance of these results would have been better understood.

Can we talk about this in the discussion? May be cite articles talking about outcomes of catheter ablation in general for atrial arrythmias?

4) Please review reviewer comments below

We look forward to receiving your revised manuscript.

Kind regards,

Vikramaditya Samala Venkata

Academic Editor

PLOS ONE

Journal Requirements:

Additional Editor Comments (if provided):

Reviewers' comments:

Reviewer's Responses to Questions

**Comments to the Author**

1. If the authors have adequately addressed your comments raised in a previous round of review and you feel that this manuscript is now acceptable for publication, you may indicate that here to bypass the “Comments to the Author” section, enter your conflict of interest statement in the “Confidential to Editor” section, and submit your "Accept" recommendation.

Reviewer #3: All comments have been addressed

Reviewer #4: All comments have been addressed

Reviewer #5: All comments have been addressed

Reviewer #6: All comments have been addressed

2. Is the manuscript technically sound, and do the data support the conclusions?

Reviewer #3: Yes

Reviewer #4: Yes

Reviewer #5: Yes

Reviewer #6: Yes

3. Has the statistical analysis been performed appropriately and rigorously? 

Reviewer #3: Yes

Reviewer #4: Yes

Reviewer #5: Yes

Reviewer #6: Yes

4. Have the authors made all data underlying the findings in their manuscript fully available?

Reviewer #3: Yes

Reviewer #4: Yes

Reviewer #5: Yes

Reviewer #6: Yes

5. Is the manuscript presented in an intelligible fashion and written in standard English?

Reviewer #3: Yes

Reviewer #4: Yes

Reviewer #5: Yes

Reviewer #6: Yes

6. Review Comments to the Author

Reviewer #3: 1) NYHA Class 2 was noted in the page 7, Table 1 baseline characteristics. In Page 10, prior to ablation NYHA Class 3 was mentioned. Was baseline NYHA >2?, please clarify.

2) Stage of Cardiac amyloidosis, when cardiac ablation was performed

3) Page 2 -Background: This study sought to investigate the long-term impact of catheter ablation in patients with CA and AA. Page 4 sentence 2,3-The aim of this study was to explore the efficacy and impact of catheter ablation in CA complicated by AA in terms of reversal of congestive heart failure and mortality.

Please clarify in page 2 background section, is the study designed to evaluate impact on cardiological outcomes?

4) Page 9 -Mortality rate 39 % (12 patients)- 3 End stage CHF ,1 Sudden Cardiac Death,1 stroke (total 5). 1 pt passed away from Covid ,1 patient passed away from unknow cause (Total 2). Another 5 patients passed away from later complication of Amyloidosis? Please clarify.

5) Page 8- AV nodal ablation and ICD, have their own impact on CHF and mortality, potential confounding factor

Reviewer #4: Well written paper and good research design. I am pleasantly surprised that the researchers were able to find so many patients with Atrial tachycardia and Cardiac Amyloidosis who underwent ablation. I would have loved to see a discussion about comparing the impacts of catheter ablation in patients with cardiac amyloidosis and patients who do not have amylodosis. This would have put the results in a better perspective. Please consider including this in the discussion portion of the paper.

Reviewer #5: Authors have addressed all the prior comments by reviewers and editors and manuscript can be accepted for publication.

Reviewer #6: Amyloidosis is a rare diagnosis and CA is especially diagnosed very late in the process. The limited study population and late referral to tertiary centers makes this a challenging endeavor. The study is provided with relevant intro, patient characteristics, statistical analyses and references. Additional studies with increased enrollment is needed. In the interim, smaller studies like these with an understanding of its limitations are always helpful to advance the understanding of this rare diagnosis.

7. PLOS authors have the option to publish the peer review history of their article (what does this mean?). If published, this will include your full peer review and any attached files.

Reviewer #3: **Yes: **Srikanth Puli

Reviewer #4: No

Reviewer #5: No

Reviewer #6: No

---

## [Author Response · Author response to Decision Letter 2]

6 Oct 2023

PONE-D-22-29848R2

Catheter Ablation of Atrial Arrhythmias in Cardiac Amyloidosis. Impact on Heart Failure and Mortality

PLOS ONE

1. There are minor grammatical errors in the abstract section. In methods section: “ AA were atrial fibrillation..” this statement needs better grammatical construction. Also in this section authors mention “ Atrial common flutter” , do you mean “ Atrial flutter” 

Answer: this has been corrected: “AA subtypes included atrial fibrillation (AFib) in 22 (paroxysmal in 10 and persistent in 12), atrial flutter (AFl) in 17 and atrial tachycardia (AT) in 11 patients.” See abstract

2. Its not clear looking at the abstract about what the main aims of the study are. In the methods section—may be we should explain that we compared outcomes before and after ablation etc? to assess long term efficacy of ablation etc

Answer: aims have been more developed. 

Aims: “This study sought to investigate the long-term efficacy and impact on HF progression of catheter ablation in patients with CA and AA.” See abstract

Methods: “Long-term AA recurrence rates were evaluated along with the impact of sinus rhythm (SR) maintenance on HF and mortality.” See abstract

Throughout rest of the article, there are statements which need grammatical correction such as

Asnwer: The manuscript has been corrected by a native English author “AA undergoing ablation were AFib in 22 (71%) (paroxysmal in 10 and persistent in 12), CTI-dependent AF in 17 (55%) and AT in 11 (35%)”---- better statement---- ‘ Among patients undergoing ablation for arrythmia, 71% had afib, 55% had CTI-dependent afib and 11% had AT” etc

This has been corrected: “All 31 patients underwent catheter ablation for AA: 22 (71%) for Afib (paroxysmal in 10 and persistent in 12), 17 (55%) for CTI-dependent AF and 11 (35%) for AT” page 7 lines 7-8

“A previous history of catheter ablation was present in only two cases (for AF)” – correct grammar “ only 2 cases had a hx of catheter ablation prior to our study.

This has been corrected: “Two patients had a history of prior catheter ablation (for AFl)” Page 7 line 10

“When comparing before and after ablation while in SR (median 9 months later, IQR 17), creatinine and natriuretic peptides levels significantly decreased as well as NYHA class, without relevant change in weight, while left atrial dimensions and pulmonary arterial pressures, although decreased, did not significantly change (see table 2)”-- Full stop after “change in weight”. Then last statement should start ( my suggestion)

This has been corrected also: “Over a median follow-up of 9 months (IQR 17), serum creatinine and natriuretic peptides levels significantly decreased with improvements in NYHA class in patients remaining in sinus rhythm, without a significant change in weight. There was a trend towards reductions in left atrial dimension and pulmonary arterial pressure after ablation in patients in sinus rhythm (see table 2)” page 9 lines 12-16

may be get assistance of professional grammatical correction services

After reviewing by an English native physician, we hope that the manuscript is now free from grammatical issues.

3. How do we know that this study shows long term efficacy of ablation in CA patients?

Is ablation in CA patients better or worse than ablation in patients without CA? Whats the control population to compare with?. How do we know that history of amyloidosis interacting/or not interacting with ablation outcomes

If we compared effects of ablation between patients who had cardiac amyloidosis and patients who did not have cardiac amyloidosis: then significance of these results would have been better understood.

Can we talk about this in the discussion? May be cite articles talking about outcomes of catheter ablation in general for atrial arrythmias?

Answer: although such comparison was not the aim of this study, a short comment about the differences between AFib ablation in CA and other settings has been added in the discussion

“This is in comparison to other clinical situations, where long-term success rates of Afib ablation range between 50% and 80% (28). Moreover, the positive impact of sinus rhythm maintenance with Afib ablation in heart failure is well established (29). Page 11 lines 17-20

Comments to the Author

Reviewer #3: 1) NYHA Class 2 was noted in the page 7, Table 1 baseline characteristics. In Page 10, prior to ablation NYHA Class 3 was mentioned. Was baseline NYHA >2?, please clarify.

Answer: median NHYA class was 2 in the whole population at baseline, but was 3 at baseline in the group of patients in whom comparisons could be made because remaining in sinus rhythm (mentioned in the legend of table 2). 

2) Stage of Cardiac amyloidosis, when cardiac ablation was performed

Answer: stage of cardiac amyloid is now mentioned in table 1

European score (AL-CA) II 2/9 IIIa 6/9 III b 1/9

NAC score (ATTR-CA) I 5/18 II 5/18 III 8/18

3) Page 2 -Background: This study sought to investigate the long-term impact of catheter ablation in patients with CA and AA. Page 4 sentence 2,3-The aim of this study was to explore the efficacy and impact of catheter ablation in CA complicated by AA in terms of reversal of congestive heart failure and mortality.

Please clarify in page 2 background section, is the study designed to evaluate impact on cardiological outcomes?

Answer: this has been changed in the abstract (see answer to the editor’s comments): 

Aims: “This study sought to investigate the long-term efficacy and impact on HF progression of catheter ablation in patients with CA and AA.” See abstract

Methods: “Long-term AA recurrence rates were evaluated along with the impact of sinus rhythm (SR) maintenance on HF and mortality.” See abstract

4) Page 9 -Mortality rate 39 % (12 patients)- 3 End stage CHF ,1 Sudden Cardiac Death,1 stroke (total 5). 1 pt passed away from Covid ,1 patient passed away from unknow cause (Total 2). Another 5 patients passed away from later complication of Amyloidosis? Please clarify.

Answer: yes 5 patients died from non-cardiac complications of amyloidosis: this bas been detailed: “while late noncardiac complications of amyloidosis were the cause of death in five patients (pneumopathy in two, degradation of general conditions in two and sepsis in one).” Page 9 lines 5-7

5) Page 8- AV nodal ablation and ICD, have their own impact on CHF and mortality, potential confounding factor

Answer: we agree although ICD did not modify mortality in this population, and previous AV node ablation did not succeed in treating heart failure despite controled heart rate. This has been added in the limitations and results.

“Presence of ICD or previous atrio-ventricular node ablation may have contributed to morbidity or mortality outcomes, however no differences in mortality between implanted and non-implanted patients were observed, and the few patients with prior atrio-ventricular node ablation manifested congestive heart failure despite ventricular rate control prior to AA ablation.” Page 13 lines 5-8

“There was no significant difference in mortality between ICD-implanted and non-implanted patients (25% vs 47%, p=ns)” page 9 lines 9-11

Reviewer #4: Well written paper and good research design. I am pleasantly surprised that the researchers were able to find so many patients with Atrial tachycardia and Cardiac Amyloidosis who underwent ablation. I would have loved to see a discussion about comparing the impacts of catheter ablation in patients with cardiac amyloidosis and patients who do not have amyloidosis. This would have put the results in a better perspective. Please consider including this in the discussion portion of the paper.

This has been done, see answer to editor’s comments above: although such comparison was not the aim of this study, a short comment about the differences between AFib ablation in CA and other settings has been added in the discussion

“This is in comparison to other clinical situations, where long-term success rates of Afib ablation range between 50% and 80% (28). Moreover, the positive impact of sinus rhythm maintenance with Afib ablation in heart failure is well established (29). Page 11 lines 17-20

Reviewer #5: Authors have addressed all the prior comments by reviewers and editors and manuscript can be accepted for publication.

We thank this reviewer

Reviewer #6: Amyloidosis is a rare diagnosis and CA is especially diagnosed very late in the process. The limited study population and late referral to tertiary centers makes this a challenging endeavor. The study is provided with relevant intro, patient characteristics, statistical analyses and references. Additional studies with increased enrollment is needed. In the interim, smaller studies like these with an understanding of its limitations are always helpful to advance the understanding of this rare diagnosis.

We thank this reviewer for these nice comments

---

## [Editor Report · Decision Letter 3]

3 Nov 2023

PONE-D-22-29848R3

Catheter Ablation of Atrial Arrhythmias in Cardiac Amyloidosis. Impact on Heart Failure and Mortality

PLOS ONE

Dear Dr. Maury,

Thank you for submitting your manuscript to PLOS ONE. After careful consideration, we have decided that your manuscript does not meet our criteria for publication and must therefore be rejected.

Specifically:

I do not completely agree with conclusions of the results as stated by authors

In the results section

When results are not significant, not sure we can there is a positive trend ( especially when the total N is also very low), So not sure if we can come to this conclusion

-Ex” Lef atrial volume and Sys pulm artery pressure: P is 0.6 and 0.26 with only minimal change in pulm artery pressure?). So not sure if we can come to this conclusion

-There was no significant difference in HF hospitalizations ( p=0.1) and all cause mortality (p=0.2)when comparing with AA recurrence.

-Even in patients with permanent AA, p is not significant ( so not sure we can say trend towards higher mortality)

So in conclusion, I agree we can say may be NYHA class and BNP numbers are improving, but rest of the parameters are not showing a significant difference, so not sure we can say positive trend repeatedly despite P value not being significant

Paper still has some grammatical errors which need to be corrected ( please see attached file with comments)In the introduction: authors mention that rate control is challenging due to negative inotropic and chronotropic effects of drugs ( can we explain more about this ?) and can we explain why only amio is the drug usedDiscussion section needs to report results of the paper more in comparison with other ( please see attached paper with comments)

Kind regards,

Vikramaditya Samala Venkata

Academic Editor

PLOS ONE

- - - - -

---

## [Author Response · Author response to Decision Letter 3]

16 Dec 2023

PONE-D-22-29848R3

Catheter Ablation of Atrial Arrhythmias in Cardiac Amyloidosis. Impact on Heart Failure and Mortality

PLOS ONE

Answers to the editor’s comments

I do not completely agree with conclusions of the results as stated by authors

In the results section

When results are not significant, not sure we can there is a positive trend ( especially when the total N is also very low), So not sure if we can come to this conclusion

-Ex” Lef atrial volume and Sys pulm artery pressure: P is 0.6 and 0.26 with only minimal change in pulm artery pressure?). So not sure if we can come to this conclusion

-There was no significant difference in HF hospitalizations ( p=0.1) and all cause mortality (p=0.2)when comparing with AA recurrence.

-Even in patients with permanent AA, p is not significant ( so not sure we can say trend towards higher mortality)

So in conclusion, I agree we can say may be NYHA class and BNP numbers are improving, but rest of the parameters are not showing a significant difference, so not sure we can say positive trend repeatedly despite P value not being significant

Answer: We agree this was too optimistic. We have changed most of these sentences

“left atrial dimension and pulmonary arterial pressure decreased after ablation in patients in sinus rhythm although not significantly”. (page 10, line 1)

“All-cause mortality was not associated with AA recurrence (5/17 wo AA recurrence vs 7/14 with AA recurrence, p=0.2), and mortality in patients with permanent AA (5/10) was not significantly higher compared to patients remaining in SR (7/21) (p=0.3)”. page 10 line 11

However we decided to keep unchanged: 

“There was a trend toward less hospitalizations for HF in patients remaining in SR over the follow-up period (2/21) compared to patients with AA recurrence (5/10) (p=0.1).” (p=0.1 may be considered as a trend in such a limited population) page 10 line 7

“All 3 patients with deaths caused by intractable HF had AA recurrence compared to 11 out of the remaining 28 long-term survivors (or dead but not related to HF) (p=0.04)”. page 10 line 9

Paper still has some grammatical errors which need to be corrected ( please see attached file with comments)

We have modified the manuscript accordingly

In the introduction: authors mention that rate control is challenging due to negative inotropic and chronotropic effects of drugs ( can we explain more about this ?) and can we explain why only amio is the drug used

This has been more explained : “Since cardiac output in CA is highly dependent on heart rate because of altered diastolic filling and since CA is associated with frequent conduction disturbances, rate control is challenging because of the vasodilator, potentially toxic, negative inotropic, dromotropic and especially chronotropic effects of drugs (2, 6). Due to before mentioned reasons, amiodarone is the only anti-arrhythmic drug used in this population, but it has been shown to be ineffective in maintaining SR in majority of the cases (10).” Page 3 line 17

Discussion section needs to report results of the paper more in comparison with other ( please see attached paper with comments)answer to the editor’s comments

A chapter dedicated to the comparison with other studies was already included rather early in the discussion, so we feel this part is enough complete and rather well placed in the discussion part

---

## [Decision Letter · Decision Letter 4]

23 Mar 2024

Catheter Ablation of Atrial Arrhythmias in Cardiac Amyloidosis. Impact on Heart Failure and Mortality

PONE-D-22-29848R4

Dear Dr. Maury,

We’re pleased to inform you that your manuscript has been judged scientifically suitable for publication and will be formally accepted for publication once it meets all outstanding technical requirements.

Kind regards,

Neil

Academic Editor

PLOS ONE

Additional Editor Comments (optional):

Reviewers' comments:

Reviewer's Responses to Questions

**Comments to the Author**

1. If the authors have adequately addressed your comments raised in a previous round of review and you feel that this manuscript is now acceptable for publication, you may indicate that here to bypass the “Comments to the Author” section, enter your conflict of interest statement in the “Confidential to Editor” section, and submit your "Accept" recommendation.

Reviewer #7: All comments have been addressed

Reviewer #8: All comments have been addressed

2. Is the manuscript technically sound, and do the data support the conclusions?

Reviewer #7: Yes

Reviewer #8: Yes

3. Has the statistical analysis been performed appropriately and rigorously? 

Reviewer #7: I Don't Know

Reviewer #8: Yes

4. Have the authors made all data underlying the findings in their manuscript fully available?

Reviewer #7: Yes

Reviewer #8: Yes

5. Is the manuscript presented in an intelligible fashion and written in standard English?

Reviewer #7: Yes

Reviewer #8: Yes

6. Review Comments to the Author

Reviewer #7: The population is extremely low and needs to be increased. It is mentioned that the facility is a tertiary facility which impacts the pool of available patients.However, this may be a good start for following studies. The P values were also addressed.

Reviewer #8: Catheter Ablation of Atrial Arrhythmia's in Cardiac Amyloid. Impact on HF and Mortality.

It's a wonderful hypothesis generating paper. The authors have tackled a difficult question in a population with multiple variables and with high short-term mortality. Any effort to improve symptoms, reduce hospitalizations and improve long term survival is commendable. It is also clear that these are relatively sicker patients referred to tertiary care facilities and therefore at high risk of morbidity and mortality despite advanced therapies.

Since the patient numbers in any of the previous and current studies is small, where AA ablation is deployed as a strategy to reduce HF hospitalization and improve symptoms, there is a clear need to establish a registry to understand such outcomes.

Behavior and natural history of AL and ATTR CA is different. AL tends to behave as an acute myocarditis, with early symptoms and rapid progression to end stage heart failure while ATTR behaves in slower amyloid deposits in atria and ventricle and conduction system (1). NSR is maintained longer in AL and Afib tends to be more common in ATTR. Once Afib is established in AL, it suggests poor prognosis (2). Onset of CHF in CA is a harbinger of poor prognosis and may not only reflect stage of the disease, but it has worse outcomes in patients with AL (3).

It is worth mentioning the pathophysiology of the disease and stage of the disease. Low EDV, reduced SV, atrial electromechanical dissociation (patient has NSR while atria show impaired contractility), increase HR and autoimmune dysfunction may all impact symptoms and progression to HF and impact survival. Risk stratification of CA and clinical management hence becomes challenging (4). Higher stage of ATTR and higher NYHA class and maintenance of NSR impacts survival (5).

Hence, variables of symptoms, hospitalization and survival include Type of CA (ATTR more prone to Afib), thromboembolism risk, stage of CA, NYHA class. Prevalence of HF in ATTR is 6.3% (1-21%) and relatively less common in AL (1.2/100thousand cases). ATTR is also associated more commonly with paradoxical low flow, low gradient AS with a prevalence of 4-29% (6).

The next step should be setting up a registry of CA patients separately for AL and ATTR types. Further stratifying them to stage of the disease and NYHA class, finding patients with AA (predominantly afib), ablating afib early before disease reaches advanced stage and looking at its impact on HF onset, recurrent hospitalization for HF and survival in a prospective fashion. This would also address the power (n) of analysis to get meaningful data. This may also lead to development of longer lasting post ablative strategies (antiarrhythmics) to maintain patients in sinus rhythm. This should go hand in hand with newer disease modifying strategies that reduce amyloid deposits in heart and elsewhere.

1. Capelli F, Cir Heart Failure 2020;13:e 006619

2. Ng PLF, Ann. Non invasive Electrophysiology 2022;27:e1267

3. Thakker, Am J Cardiol 2021;143:125-130

4. Laptsera N, J. Clinical Med, 2023;12:2581

5. JACC Clin Electrophysiology 2020;6:1118-1127

6. Ternacle J, JACC 2019;74(21):2638-2651

7. PLOS authors have the option to publish the peer review history of their article (what does this mean?). If published, this will include your full peer review and any attached files.

Reviewer #7: **Yes: **Maha Ahmed

Reviewer #8: **Yes: **Nadeem Afridi

---

## [Editor Report · Acceptance letter]

27 Mar 2024

PONE-D-22-29848R4 

PLOS ONE

Dear Dr. Maury, 

I'm pleased to inform you that your manuscript has been deemed suitable for publication in PLOS ONE. Congratulations! Your manuscript is now being handed over to our production team.

Kind regards, 

on behalf of

Dr. Neil Patel 

Academic Editor

PLOS ONE